# Material Security Scale as a Measurement of Poverty among Key Populations At-Risk for HIV/AIDS in Malaysia: An Implication for People Who Use Drugs and Transgender People during the COVID-19 Pandemic

**DOI:** 10.3390/ijerph19158997

**Published:** 2022-07-24

**Authors:** Nur Afiqah Mohd Salleh, Ahsan Ahmad, Balasingam Vicknasingam, Adeeba Kamarulzaman, 'Abqariyah Yahya

**Affiliations:** 1Department of Social and Preventive Medicine, Faculty of Medicine, University of Malaya, Kuala Lumpur 50603, Malaysia; imohdsalleh@um.edu.my; 2Centre of Excellence of Research in AIDS, Faculty of Medicine, University of Malaya, Kuala Lumpur 50603, Malaysia; ahsan.ahmad@yale.edu (A.A.); adeeba@ummc.edu.my (A.K.); 3Yale School of Medicine, Yale University, New Heven, CT 06510, USA; 4Centre for Drug Research, Universiti Sains Malaysia, Gelugor 11800, Malaysia; vickna@usm.my; 5Department of Medicine, Faculty of Medicine, University of Malaya, Kuala Lumpur 50603, Malaysia

**Keywords:** HIV, social determinants of health, poverty, reliability, validity

## Abstract

The HIV epidemic is fueled by poverty; yet, methods to measure poverty remain scarce among populations at risk for HIV infection and disease progression to AIDS in Malaysia. Between August and November 2020, using data from a cross-sectional study of people who use drugs, (PWUD), transgender people, sex workers and men who have sex with men, this study examined the reliability and validity of a material security scale as a measurement of poverty. Additionally, we assessed factors associated with material security scores. We performed confirmatory factor analysis (CFA) for 268 study participants included in the analysis. A revised nine-item three-factor structure of the material security scale demonstrated an excellent fit in CFA. The revised material security score displayed good reliability, with Cronbach’s alpha of 0.843, 0.826 and 0.818 for housing, economic resources and basic needs factors, respectively. In a subsequent analysis, PWUD and transgender people were less likely to present good material security scores during the pandemic, compared to their counterparts. The revised nine-item scale is a useful tool to assess poverty among key populations at-risk for HIV/AIDS with the potential to be extrapolated in similar income settings.

## 1. Introduction

Four decades into the HIV epidemic, it is established that poverty is strongly associated with high levels of HIV-related morbidity and mortality through various mechanisms. Socioeconomic marginalization, for example, may prevent an individual from having adequate resources to access healthcare and achieve optimal treatment outcomes due to competing priorities [1,2]. The impact of poverty on health is further exacerbated by high levels of stigma and discrimination toward people living with HIV and the risky behaviors associated with HIV transmission [3]. Specifically, the criminalization of drug possession for personal use, sexual orientation and sex work may render these sub-groups more vulnerable to limited employment and development opportunities due to criminal records.

Measurements of poverty in relation to health vary across studies, with income-based measures including individual monthly income and average household income dominating the conceptualization of poverty in the literature [4]. However, income-based measures may be limited by the geographic variation of cost of living and the risk of underreporting because other sources of revenue, i.e., welfare assistance, are not commonly included in the measurement [5,6]. In addition to income-based measures, other analyses have included neighborhood characteristics, food security and other structural inequalities to capture poverty and its association with HIV care outcomes [7,8]. Despite the advantage of reporting well-defined indicators of poverty, these single-domain measures are not inclusive of other forms of material possessions that may collectively influence day-to-day hardships.

The material security scale has been modified from the Family Resource Scale (FRS) which was previously used to measure the adequacy of resources in households with young children [9]. Encompassing individual daily necessities, the material security scale commonly identifies housing, food and economic resources which collectively move beyond income in capturing daily hardships. Previous studies that assessed material security among key at-risk populations have modified the FRS scale by reducing 30 items to 18 items, including items related to child-specific resources, to reduce the possibility of missing data [10]. Specifically, studies in New York, USA have examined material resources among people at risk of acquiring HIV including people who use drugs (PWUD) and men who have sex with men (MSM) using the 18-item material security scale with reports of good validity and reliability scores [10,11]. The same scale has additionally been demonstrated to be valid and reliable in studies in Vancouver, Canada for assessing the association of material security with exposure to violence and adherence to HIV treatment among PWUD [12,13]. Although the material security scale offers a robust measure of poverty, the scale has yet to be adapted and validated in low and middle-income countries, particularly in southeast Asia.

In Malaysia, the HIV epidemic is concentrated among at-risk populations including PWUD, MSM, transgender people and sex workers, with new infections in the last few years now being attributed to sexual transmission [14]. Markers of poverty, including income levels below the poverty line and challenges in finding employment, have been commonly reported by key at-risk populations for HIV/AIDS in the country [15,16,17]. However, studies that explore measurements of poverty among at-risk populations in the context of HIV/AIDS remain scarce, with limited knowledge on the quantification of material resources. The lack of validated and reliable measures hampers public health efforts to effectively advance the design and implementation of holistic, person-centered care models for at-risk populations. Therefore, this study sought to measure if the material security scale provides valid and reliable evidence of material resources among key populations at-risk for HIV/AIDS in Malaysia. Additionally, this study aimed to assess factors associated with poor material security scores before and during the COVID-19 pandemic to allow a better understanding of key groups that were more likely to be vulnerable during the pandemic.

## 2. Materials and Methods

### 2.1. Study Population

This cross-sectional study was conducted between August–November 2020 to assess the overall well-being of at-risk populations including PWUD, MSM, transgender people and sex workers, before and during the COVID-19 pandemic. Study participants were recruited through ten community-based organizations (CBO) that provide HIV-related services across Malaysia. Eligibility criteria included being at least 18 years old, registered as a client at selected CBOs for at least 6 months and being able to provide consent. Each CBO provided the number of individuals within each key population group (PWUD, MSM, transgender people and sex workers) who registered at their respective organizations. A number of 6439 individuals were active clients registered in the ten community-based organizations, regardless of HIV status. The sample size was 250 participants across all four key population groups from these CBOs. The number of samples required from each CBO is calculated based on this sample size using proportionate stratified sampling.

Due to COVID-19 movement control restrictions, questionnaires were administered by trained interviewers through either telephone calls or in-person. Participants were referred by outreach workers via telephone calls or recruited physically at CBOS. Data was captured using Research Electronic Data Capture (REDCap), a secure web application for building and managing online surveys and databases. Study participants were remunerated RM30 (approximately USD 7) upon questionnaire completion.

### 2.2. Ethical Approval

Ethical approval was obtained from the Medical Research Ethics Committee, University Malaya Medical Center (MRECID.NO: 202053-85880).

### 2.3. Study Instruments

The questionnaire consists of seven sections on sociodemographic, material security, illicit drug use, sexual behaviors, the criminal justice system, access to healthcare and HIV-related services, such as the needle and syringe exchange program, and opioid agonist therapy. Participants were asked whether they had adequate access to a series of material resources using the material security scale, before and during the COVID-19 pandemic, as the following: (1) Food for two meals a day; (2) A house or apartment; (3) Indoor plumbing and water; (4) Money to buy necessities, (5) Enough clothes for yourself (and family); (6) Money to pay monthly bills; (7) A job for yourself (or your partner); (8) Access to medical care for yourself and family; (9) Social Assistance; (10) Time to get enough sleep/rest; (11) Access to a telephone; (12) Money to save; (13) Money to spend on self; (14) Dependable transportation; (15) Money for entertainment; (16) Dental care for self; (17) Adequate furniture and (18) Heat for housing (Table 1). Responses were based on a five-point Likert scale: 1, never; 2, occasionally; 3, sometimes; 4, usually; and 5, always having access. Higher scores indicate higher levels of material security.

The standard forward-backward translation technique was used to translate the English version of the material security scale into Malay. The 18-item material security scale has been reduced to 14-items for suitability in the local context. Based on the feedback from a panel of experts that comprised of public health professionals and community healthcare workers, the four items that were removed were “money for entertainment”, “dental care for self”, “adequate furniture” and “heat for housing”. Items that were removed were not relevant to the local context and/or did not reflect basic material resources needed during a pandemic. Based on 10 cases per item, a minimum sample size of 140 participants was needed for factor analysis for questionnaire validation [18]. A greater number of participants were recruited as a larger sample size produces a more stable and reliable factor structure [19].

### 2.4. Statistical Analysis

#### 2.4.1. Construct Validity of the Material Security Scale Using Confirmatory Factor Analysis

Validity and reliability analyses were restricted to all observations (268 participants) before the pandemic, with no missing values for any of the 14-items. A confirmatory factor analysis (CFA) was performed to determine whether the items conformed to the three-factor structure that was expected based on a pre-established theory from previous studies. In analyzing CFA, three types of validity measurements were assessed. First, construct validity was performed. Fitness indices such as Root-Mean-Square Error of Approximation (RMSEA); Comparative Fit Index (CFI); Normed Fit Index (NFI); Tucker–Lewis Index (TLI); Chisq/df and Akaike Information Criteria (AIC) were used to assess the construct validity. A full model including all items was constructed. Items with factor loadings of <0.6 and R^2^ < 0.4 were removed at each reduced model. This iterative process was continued until all items remained had factor loadings of >0.6 and R^2^ > 0.4. The final model was selected based on acceptable fit indices, including RMSEA < 0.08, CFI > 0.90, NFI > 0.90, TLI > 0.90, Chisq/df < 3.0 and the lowest AIC score. A multivariate normality check was performed on the final model by assessing Mahalanobis distance, skewness and Kurtosis index. Additionally, a Bollen–Stine Bootstrap method was considered based on the result of the normality assessment.

#### 2.4.2. Discriminant Validity and Convergence Validity of the Items within Constructs

Discriminant validity was used to assess the extent to which conceptually similar items are distinct from one another. The average variance extracted (AVE) and composite reliability (CR) were calculated with an acceptable fit of at least 0.5 and 0.6, respectively. Finally, convergence validity was used to assess the extent to which conceptually similar items are related to one another. To assess this, square root of AVE (discriminant validity) for each factor was calculated and compared with correlation values between factors. The CFA was performed using IBM SPSS and AMOS version 27 (Amos Development Corp, Meadville, PA, USA).

#### 2.4.3. Reliability of the Material Security Scale

The internal consistency of the material security scale that demonstrated the extent to which items were measuring the same construct was assessed using Cronbach’s alpha coefficient. A Cronbach’s alpha of 0.7 was deemed acceptable. The analysis of the revised material security scale, based on the final model, was performed using IBM SPSS Statistic software version 27 (SPSS Inc., Chicago, IL, USA).

#### 2.4.4. Factors Associated with Poor Material Security Scores before and during the COVID-19 Pandemic

In a subsequent analysis to explore factors associated with material security score during the COVID-19 pandemic among all 292 study participants, the median material score before the pandemic was used as a cut-off point. We considered a range of variables including age (per 10 years older); education, as determined by completion of high school or above (yes vs. no); ethnicity (Malays vs. non-Malays); residency (Central Region vs. non-Central region); marital status (married vs. non-married); income during the pandemic (per RM100 increase) and HIV status (yes vs. no). We also considered variables that described key populations at-risk for HIV/AIDS including gender (male, female and transgender); engagement in sex work (yes vs. no); people who use drugs, as determined by any illicit drug use in the past one year (yes vs. no) and MSM, as determined by men who self-identified as non-heterosexual (yes vs. no).

Using a generalized logistic regression, we constructed multivariable models of factors associated with material security scores before and during the pandemic, dichotomized as poor and good, as determined by the median material security score before the pandemic. Using an a priori-defined modeling procedure, we fit a full model that included all variables with *p*-values less than 0.10 in bivariable analyses. After noting the Akaike Information Criterion (AIC) values for the full model, we constructed reduced models by using a backward selection approach, sequentially eliminating the variable that was associated with the largest *p*-value. We continued this process until zero variables were left in the model. Using this backwards-selection procedure, we constructed the final model with the best fit, as indicated by the lowest AIC value. Analyses were performed by using R statistical software Version 1.0.143 (R Core Team, Vienna, Austria). 

## 3. Results

### 3.1. Sociodemographic and Behavioral Characteristics

A total of 292 participants were included in the study sample for the reliability and validity analysis, of which 145 (50%) were male, 74 (25%) were female and 71 (24%) were transgender, as demonstrated in Table 1. Key populations included 77 (26%) PWUD; 71 (24%) MSM and 120 (41%) sex workers. The median age was 37 years old with the majority of the participants in the 31–40 age group. A total number of 221 (76%) were Malays, 31 (11%) were Native Sabah and Sarawak, 18 (6%) were Chinese, and 17 (5%) were Indians.

### 3.2. Construct Validity

In CFA, eight items were placed under Factor 1 (Basic Needs); four items were placed under Factor 2 (Economic) and two items were placed under Factor 3 (Housing). In the first model, fit indices were not acceptable, with three items demonstrating factor loadings and R^2^ values below the threshold levels (i.e., factor loadings of <0.6 and R^2^ < 0.4). Specifically, items “social assistance”, “access to dependable transportation” and “money to save”, were removed due to low factor loadings. Subsequently, in the second model, although factor loadings of the remaining 11 items were fairly good, the fit indices were not acceptable. Additionally, in performing discriminant validity, the modification indices (MI) between error variances were more than 15, which suggested redundant items in two distinct factors. Specifically, items “money to pay monthly bills” and “time to get enough sleep or rest” were explained by Factor 2 (Economic). Item “time to get enough sleep or rest” was removed due to unmet requirements for factor loading and R^2^. In the subsequent third model, item “access to a telephone to at least make a phone call” demonstrated factor loading <0.60 and was also removed. Improvement in model fit indices was demonstrated after the removal of these five items.

In the final model, the revised three-factor structure of the material security scale (Figure 1), with only nine items and one pair of error covariances was found to be acceptable (RMSEA = 0.115, CFI = 0.939, NFI = 0.923, TLI = 0.904 and Chi sq/df = 4.535). Furthermore, the final model demonstrated an AIC value of 166.302; the lowest AIC value when compared to AIC values of other models. The value of Mahalanobis distance was 91.57, which was above the threshold level of 90, indicating the presence of outliers. Additionally, assessment of normality demonstrated composite reliability (CR) for Skewness Index and Kurtosis values exceeded the absolute values of 8.0 and 3.0, respectively. Given that the assumption of multivariate normality was not met, a Bollen–Stine test with 1000 bootstraps was performed. In Bollen–Stine bootstrapping, the final model fitted better in 999 bootstrap samples while failing to fit in only one bootstrap sample, with a significant *p*-value of 0.002. Therefore, the model fit was considered acceptable.

### 3.3. Discriminant and Convergence Validity

Discriminant validity demonstrated acceptable values of AVE and CR of more than 0.5 and 0.6, respectively. AVE values for Factor 1 (Basic Needs), Factor 2 (Economic) and Factor 3 (Housing) were 0.534, 0.622 and 0.779, respectively. Meanwhile, CR values for Factor 1 (Basic Needs), Factor 2 (Economic) and Factor 3 (Housing) were 0.818, 0.831 and 0.875, respectively. Values of AVE, CR and factor loadings of each item are demonstrated in Table 2.

Convergence validity demonstrated acceptable correlation values between constructs that were below the threshold level of 0.85. Specifically, the correlation between Factor 1 (Basic Needs) and Factor 2 (Economic) was 0.87; the correlation between Factor 1 (Basic Needs) and Factor 3 (Housing) was 0.73 and the correlation between Factor 2 (Economic) and Factor 3 (Housing) was 0.40. As demonstrated in Table 3, the square root of AVE for each factor was presented in a bolded and diagonal value; meanwhile, other values that were not bolded represent correlation values between factors. The square root of AVE (discriminant validity) for all bolded, diagonal values of each factor was higher than correlation values between factors (not bolded), therefore demonstrating that discriminant validity for all constructs was achieved.

### 3.4. Reliability Testing

The revised nine-item Material Security displayed good internal consistency with Cronbach’s alpha coefficients of 0.818, 0.826 and 0.843 for Factor 1 (Basic Needs), Factor 2 (Economic) and Factor 3 (Housing), respectively. The total scale demonstrated a Cronbach’s alpha coefficient of 0.885.

### 3.5. Factors Associated with Good Material Security Scores before the COVID-19 Pandemic

Table 4 shows the crude and adjusted estimates of the odds of presenting good material security scores during the COVID-19 pandemic. Individuals who presented higher odds of good material security scores were those with higher income levels [(Adjusted Odds Ratio (AOR) = 1.15, 95% Confidence Interval (CI): 1.09–1.22)]. Meanwhile, individuals who were older (AOR = 0.74, 95% CI: 0.55–0.98) presented lower odds of good material security scores.

### 3.6. Factors Associated with Good Material Security Scores during the COVID-19 Pandemic

Table 5 shows the crude and adjusted estimates of the odds of presenting good material security scores during the COVID-19 pandemic. Individuals who presented higher odds of good material security scores were those with higher income levels during the pandemic (AOR = 1.07, 95% CI: 1.03–1.11). PWUD (AOR = 0.17, 95% CI: 0.04–0.54). Transgender people (AOR = 0.24, 95% CI: 0.05–0.90) presented lower odds of good material security scores compared to individuals who did not use drugs and male counterparts, respectively.

## 4. Discussion

Through confirmatory factor analysis, assessing construct, discriminant and convergence validity and reliability testing, as shown in Table 2 and Table 3, the revised nine-item material security scale is a valid and reliable tool to assess material security among key populations at-risk for HIV/AIDS in Malaysia. The application of the revised nine-item materials security scale indicates that Factor 1 (Basic Needs) which includes food, clothes, money and healthcare, has a strong correlation (0.87) with Factor 2 (Economic) which includes money and employment.

Findings from this study support a well-studied relationship between income security and better health outcomes, while also supporting previous research on how access to food, clothing and support for mental and physical health [20]. Additionally, a global review of 134 studies amidst the COVID-19 pandemic concludes that scaling up and diversifying the range of income security interventions is crucial for improving health outcomes, particularly in reducing the risk of transmitting infectious diseases [21]. Furthermore, the tool also indicates that key populations who experience criminalization of drug possession for personal use, sexual orientation and/or sex work, may also experience hardships to obtain sustainable revenue streams to sustain livelihood.

The HIV epidemic in Malaysia has evolved over the past 20 years; however, it has always been highly concentrated in socially marginalized and disenfranchised communities. Structural and economic conditions of key at-risk populations including environmental resources, constraints and access to care are influenced by stigma and discrimination [22]. To delve into this topic, acknowledging poverty as a multi-dimensional phenomenon that goes beyond income or resources to encompass the psychological pain of being poor, low achievements in education and health and a sense of vulnerability to external events, such as the COVID-19 pandemic, can inform priorities within a country’s resourcing capability.

Subsequent analyses in this present study have demonstrated that PWUD presented poor material security scores during the pandemic where PWUD had an 83% lower likelihood of presenting good material security scores compared to individuals who did not use drugs in the past one year. In addition to PWUD, transgender people were 76% less likely to present good material security scores compared to cis-males during the pandemic. Findings from this study echoed several other studies that demonstrated that the pandemic has disproportionally impacted people who have traditionally experienced preexisting challenges associated with poverty including social isolation and stigma [23,24,25]. Poverty can be further exacerbated by the lack of financial assistance in many developing countries, including those with large PWUD populations in Eastern Europe or East and South-East Asia [24].

Restrictions of movement during the pandemic as posed by the government have potentially limited the movement of PWUD, which is typically a mobile population, due to employment opportunities, having multiple residences including those that provide support and locations of drug markets [25]. Moreover, PWUD tend to work odd jobs and daily paid work (i.e., formal and informal income generating activities), therefore posing a challenge for this population to move freely during lockdowns, and in turn, to survive economically [26]. High levels of discrimination that are shaped by the criminalization of drug use and being transgender can also instill fear among these key populations, preventing them from traveling especially when the number of police roadblocks increased during lockdowns. Closures and reductions in the operational capacity of local service organizations offering critical health and social services specific for key populations, alongside disproportionate employment disruptions, may contribute towards presenting poor material security scores.

Evidence from this study points to the possibility of considering material resources to better understand the relationship between poverty and health, as opposed to the traditional view of income as a measurement of poverty. The material security scale reveals specific challenges experienced by people at risk for HIV/AIDS in navigating daily life. Findings from this study demonstrate that all three domains of material security collectively should be accessible by PWUD and transgender groups, especially during a pandemic when these groups are more likely to be impacted by material insecurity compared to their counterparts. Such findings can be useful for service providers, particularly CBOs that deliver HIV prevention and treatment services to key populations, in order to recognize and identify service priorities for groups that are at higher risks of poverty during such a pandemic. Moreover, the nine-item offers a rapid assessment of needs that can be potentially utilized by CBOs in the future. As such, the design of health programs should take into consideration the challenges of obtaining material resources during a pandemic. Future research to assess whether variation in material deprivation across the three factors is associated with high-risk behaviors, morbidity and mortality is needed.

This study has several limitations to note. First, as with all observational studies, it is not possible to draw causal inferences between exposures and outcome of interest, that is, material security. Second, the sample used was limited by the exclusive enrollment of key populations who were clients of HIV-service organizations. Therefore, findings may not be generalized to all key populations in the country as there are some who are not accessing these HIV-service organizations. Third, measurements of material security before the COVID-19 pandemic were self-reported during the pandemic, therefore they are susceptible to recall and desirability bias. Fourth, while existing studies among the same populations support the three-factor structure and the items for each factor, the three factors found in this study may not necessarily cover material resources that are truly needed by the key population in the local context. Additional validated and reliable measures pertaining to absolute versus relative poverty indicators, money-metric measures and welfare indicators can augment the nine-item material security scale. Despite these limitations, this study sheds important light on the potential impact of COVID-19 on the material well-being of key populations at risk for HIV/AIDS in Malaysia by using good reliability and validity scores of the material security scale.

## 5. Conclusions

As methods for defining and measuring poverty for key populations at risk for HIV/AIDS remain limited in low- and middle-income countries, the findings from the nine-item material security scale reinforce the need for further investigation on the role of health and social services in improving material security in Malaysia for PWUD and transgenders, especially during a pandemic.

The revised nine-item Material Security Scale is a valid instrument and can be reliably used to assess perceived poverty among key populations at-risk for HIV/AIDS in Malaysia. In particular, this tool provides a timesaving and inexpensive assessment tool for researchers to identify groups with poor material resources in countries similar to Malaysia to design targeted interventions.

## Figures and Tables

**Figure 1 ijerph-19-08997-f001:**
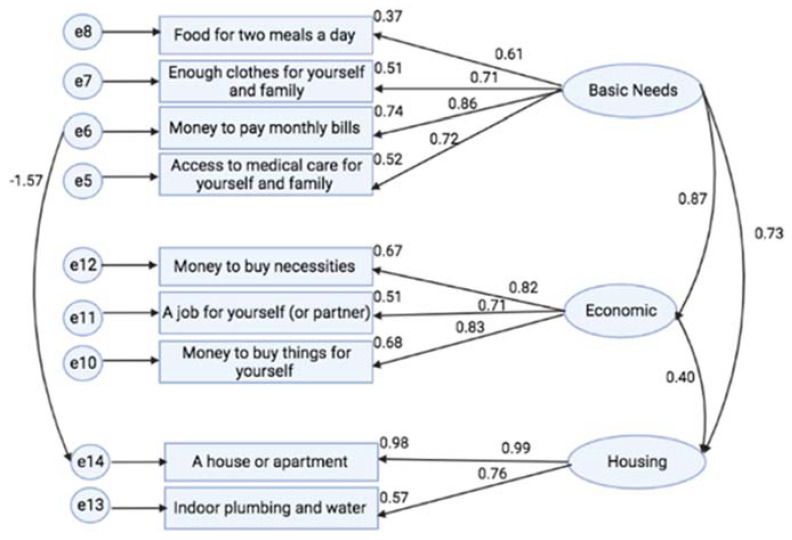
Path diagram of the revised three-factor structure of the material security scale.

**Table 1 ijerph-19-08997-t001:** Characteristics of key populations at-risk for HIV/AIDS in Malaysia, n = 292.

Characteristics	n (%)
Sex	
Male	145 (50)
Female	74 (25)
Transgender women	71 (24)
Residency	
Central region	30 (10)
Northern region	104 (36)
Southern region	42 (14)
East Coast Region	93 (32)
Sabah	22 (8)
Age	
<20 years old	5 (2)
21–30 years old	86 (29)
31–40 years old	97 (33)
41–50 years old	66 (23)
>50 years old	35 (12)
Race	
Malay	221 (76)
Chinese	18 (6)
Indian	17 (6)
Native Sabah/Sarawak	31 (11)
Education	
No formal education	8 (3)
Completed primary school	46 (16)
Completed high school	174 (60)
College/University	63 (22)
Income per month (median)	RM1200
HIV Status	
Yes	30 (10)
No	262 (90)
Men who have sex with men (MSM)	
Yes	71 (24)
No	211 (72)
Sex workers	
Yes	120 (41)
No	156 (53)
People who use drugs (PWUD)	
Yes	77 (26)
No	198 (68)

**Table 2 ijerph-19-08997-t002:** Factors loadings, average variance extracted (AVE) and composite reliability (CR) for all 9-items of the Material Security Scale.

Construct	Item	Factor Loading	AVE	CR
Housing	A house or apartment	0.99	0.779	0.874
	Indoor plumbing and water	0.76		
Economic	Money to buy necessities	0.82	0.622	0.831
	A job for yourself (or partner)	0.71		
	Money to buy things for yourself	0.83		
Basic needs	Food for two meals a day	0.61	0.534	0.818
	Enough clothes for yourself and family	0.71		
	Money to pay monthly bills	0.86		
	Access to medical care for yourself and family	0.72		

**Table 3 ijerph-19-08997-t003:** Discriminant validity for all constructs comparing square root of AVE and correlation values between factors.

	Housing	Economic	Basic Needs
Housing	0.882 *		
Economic	0.400	0.789 *	
Basic Needs	0.7300	0.87	0.731 *

* Values are the square root of AVE (discriminant validity) of each factor.

**Table 4 ijerph-19-08997-t004:** Bivariate and multivariate factors associated with good material security scores (≥32) before the COVID-19 pandemic among 292 key populations at-risk for HIV/AIDS.

	Crude		Adjusted	
Characteristic	Odds Ratio (95% CI)	*p-*Value	Odds Ratio (95% CI)	*p-*Value
Gender				
Male	Ref			
Female	0.80 (0.45–1.44)	0.4660		
Transgender	1.48 (0.83–2.66)	0.1870		
Age				
Per 10 years	0.65 (0.50–0.82)	<0.001	0.74 (0.55–0.98)	0.0479
Education				
(≥higher vs. <high school)	3.26 (1.68–6.67)	<0.001		
Residency				
(Central vs. non-Central)	0.57 (0.15–1.26)	0.171		
Ethnicity				
(Malays vs. non-Malays)	0.63 (0.34–1.13)	0.121		
Income				
Per RM100	1.13 (1.09–1.19)	<0.001	1.15 (1.09–1.22)	<0.001
Marital status				
(married vs. non-married)	0.60 (0.30–1.18)	0.144		
HIV status				
(yes vs. no)	0.85 (0.38–1.83)	0.673		
MSM				
(yes vs. no)	2.20 (1.28–3.84)	0.005		
PWUD				
(yes vs. no)	0.27 (0.15–0.48)	<0.001		
Sex worker				
(yes vs. no)	1.81 (1.11–2.99)	0.0182		

**Table 5 ijerph-19-08997-t005:** Bivariate and multivariate factors associated with good material security scores (≥32) during the COVID-19 pandemic among 292 key populations at-risk for HIV/AIDS.

	Crude	Adjusted
Characteristic	Odds Ratio (95% CI)	*p-*Value	Odds Ratio (95% CI)	*p-*Value
Gender				
Male	Ref			
Female	0.34 (0.13–0.77)	0.0154	0.49 (0.11–1.99)	0.331
Transgender	0.24 (0.08–0.59)	0.0046	0.24 (0.05–0.90)	0.047
Age				
Per 10 years	0.68 (0.47–0.95)	0.0284		
Education				
(≥ higher vs. <high school)	3.56 (1.22–15.13)	0.0408		
Residency				
(Central vs. non-Central)	0.66 (0.26–1.89)	0.398		
Ethnicity				
(Malays vs. non-Malays)	0.43 (0.14–1.05)	0.0880		
Income				
Per RM100	1.09 (1.05–1.13)	<0.001	1.07 (1.03–1.11)	<0.001
Marital status				
(married vs. non-married)	0.57 (0.31–0.93)	0.0403		
HIV status				
(yes vs. no)	0.65 (0.15–1.99)	0.502		
MSM				
(yes vs. no)	3.17 (1.60–6.27)	<0.001		
PWUD				
(yes vs. no)	0.44 (0.17–0.98)	0.061	0.17 (0.04–0.54)	0.005
Sex worker				
(yes vs. no)	0.40 (0.18–0.81)	0.0147

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
