# Peer review of "Material Security Scale as a Measurement of Poverty among Key Populations At-Risk for HIV/AIDS in Malaysia: An Implication for People Who Use Drugs and Transgender People during the COVID-19 Pandemic"

_ijerph, 2022, doi:10.3390/ijerph19158997_

Round 1

Reviewer 1 Report

This paper is very well written, very clear in its presentation. The topic is relevant and timely, and interesting.

The combination of data obtained from self-administration and interviewer administered needs at least a mention, with at minimum a look at whether there are significant differences in the data based on mode.

Regarding the methods, for future research, I encourage you to look into best practices in translation of measurements. Back translation is no longer considered best practice. Team or committee approaches as implementations of Harkness' (2002) TRAPD model (Translation-Review-Assessment-Pretesting-Documentation) are best practice. Back translation is falling in disuse for strong methodological reasons.

For additional strength in future studies, I recommend adding a qualitative look at construct validity, through cognitive interviews. 

I found only one typo. On line 275, the verb supports should be 'support'.

Reviewer 2 Report

I am grateful to the authors for their hard work and careful attention to detail on developing this manuscript. My perspective is that of a biostatistician trained in measure development.  Points for consideration are as follows:

(1) Technical merit is a strength of this paper. It is my belief that the paper is too technical and too detailed for the readership of IJERPH.  I am not worried about the sophistication of the ideas, but, rather, dragging the readers through every single decision that is not always necessary.  My recommendation at this point is to not drag readers through every single technical justification but, rather, to get to the point more directly--what was done and why. 

(2) A weakness of the paper for me was establishing a clear need for this study.  I get that measures like the one here have been used primarily in developed countries, and that the scale can be used to measure the impact of poverty among selected groups in Malaysia.  But, how does the information obtained here translate to improved quality of life for the participants? What are the actionable elements that can result from this work?  The paper appears to be silent in this regard--or, I am not appreciating their key points.  As currently written, the paper strikes me as an academic exercise without direct benefit to the communities the authors serve.  

Statistically, the paper is pretty traditional in scope and conditioned on the available data, rather than on what is needed for this to work (see point above). The paper is a fairly straightforward presentation of latent variable and linear modeling but it was unclear to me why the different steps were needed. For example, do the three latent areas cover the ones needed by this population--and, are some areas left out of consideration?  There is relative imbalance among the three factors wrt the items that make them up.  Is this by design?  Or, is that what was available? 

Based on the purpose of the study, it was clear why the authors pursued a factor analysis.  However, it was not immediately clear why the authors jumped right into CFA.  Did they do an analysis on a training sample first and then apply their methods to the remaining part of the sample?  It was also not immediately clear why they pursued bivariate and multivariate associations as well as the generalized linear (logistic) modeling.  This should be detailed out more fully in the data analysis section in subsequent revisions of the paper.  One recommendation is for the authors to craft a focused, well-designed data analysis section in which the authors detail out what they are doing in a step by step fashion along with rationale for these analyses.  I am suspecting that they will realize that not all analyses are needed to achieve their intended aims.     

Reviewer 3 Report

Review of manuscript entilted: “Material security scale as a measurement of poverty among key populations at-risk for HIV/AIDS in Malaysia: An implication for people who use drugs and transgenders during the COVID-19 pandemic” authored by NA Mohd Salleh, Ahsan Ahmad, Balasingam Vicknasingam, Adeeba Kamarulzaman and 'Abqariyah Yahya

At the beginning I want to thank you for opportunity to review this interesting manuscript.

In the presented manuscript authors, tried to create a tool for poverty assessment for populations, which are particularly prone for HIV infection. Introduction provides sufficient information for undertaking this problem. Materials and methods section needs to be rewritten to be more reader-friendly. Results are presented nicely, however some improvements have to be made. Discussion and conclusions are written logically and based on obtained results.

Overall, manuscript is good but in my opinion some improvements need to be done, below you will find my remarks.

Major concerns:

  • Materials and methods section is a little bit unreadable in the present form. I would suggest to divide it into subsections (e.g. sample, ethical approval, questionnaire, statistical analysis etc.)
  • Moreover authors stated that “(…)pandemic among all 292 study participants,(…)”, although I do not see characteristics of this population like I do for the first “dataset”. Was it totally different than the first population or some of the observations were included in the bigger population?
  • Table 4. is cut in my version of the manuscript
  • I do not quite understand pattern of showing adjusted odds ratio in your tables. Sometimes it is shown even for not-significant factors and sometimes it is hidden. If I missed explanation for this please forgive me

Minor concerns:

  • Table 1.
    • Missing values for education
    • “Sex” appears as it was n=133, I understand that this number should be for males

Round 2

Reviewer 2 Report

Still wordy and lacking in transitions and rationale for key analyses.  Lacking in off-diagonal loadings for CFA--but, I don't want to hold it up.  Of value to readership in general.  Thanks for sharing.  

Author Response

We thank Reviewer 2 for the comment and the full support of this manuscript. 

Reviewer 3 Report

Authors responded to all my concerns. Thank you and congratulations!

Author Response

Thank you Reviewer 3 for the comment and full support.